# The Effect of Preprocedural Low-Dose Ketamine for Pain and Anxiety in Patients during Thoracic Epidural Catheterization

**DOI:** 10.3390/medicina60040679

**Published:** 2024-04-22

**Authors:** Onur Küçük, Esra Sarı, Musa Zengin, Gülay Ülger, Fatma Öztürk Yalçın, Ramazan Baldemir, Mehtap Tunç, Ali Alagöz

**Affiliations:** 1Department of Anesthesiology and Reanimation, Ankara Atatürk Sanatoryum Training and Research Hospital, University of Health Sciences, 06290 Ankara, Turkey; dr.okucuk@gmail.com (O.K.); baldemir23@yahoo.com (R.B.);; 2Department of Anesthesiology and Reanimation, Ankara Etlik City Hospital, Ministry of Health, 06170 Ankara, Turkey

**Keywords:** ketamine, pain, thoracic epidural catheterization, anxiety, preprocedural low dose

## Abstract

*Background and Objectives*: Thoracic epidural catheterization (TEC) can be both uncomfortable and fearful for patients when performed awake with the thought that the procedure may be painful. The aim of this study was to assess the effect of low-dose intravenous ketamine administration on pain and anxiety during the TEC procedure. *Materials and Methods*: Sixty patients were randomly divided into two groups to receive intravenous (IV) placebo (Group P) and IV low-dose (0.15 mg/kg) ketamine (LDK) (Group K) 3 min before the procedure in a double-blind manner. A visual analog scale (VAS) was used to measure anxiety (VAS-A) and pain (VAS-P) scores. Vital parameters were monitored before premedication (T1), 20 min after premedication (T2), during skin anesthesia (T3), during TEC (T4), and 5 min after TEC (T5). VAS-A values were recorded at T1, T3, T4, and T5 periods, and VAS-P levels were noted at T3, T4, and T5 periods. *Results*: During TEC (T4), both VAS-P and VAS-A were significantly lower in Group K (*p* < 0.001). The mean VAS-A value was 10.6 mm lower, and the mean VAS-P value was 9 mm lower in Group K than in Group P at the T4 time point. Additionally, the mean VAS-P value was 7.7 mm lower in Group K compared to Group P at the T3 time point (*p* < 0.001). Both groups showed a statistically significant difference in VAS-A measurements when compared at their respective time points (*p* < 0.001). However, only Group P demonstrated a statistically significant difference in VAS-P measurements (*p* < 0.001). VAS-P values remained stable in Group K. The number of patients who did not recall the procedure was significantly higher in Group K (*p* < 0.001). Furthermore, the number of patients who would consent to the same procedure in the future was significantly higher in Group K (*p* = 0.007). *Conclusions*: A preprocedural LDK (0.15 mg/kg) can effectively prevent anxiety and pain experienced by patients during the TEC procedure. Administration of LDK may provide a more comfortable procedure process without causing ketamine-induced side effects (hemodynamic, respiratory, and psychological).

## 1. Introduction

Thoracic epidural catheterization (TEC) is considered a gold standard method of pain management for upper abdominal and thoracic surgery [1]. However, this procedure can be both unsettling and anxiety-provoking for patients when performed awake. This unfavorable experience may adversely affect the process by causing an increase in anxiety and pain [2]. Several sedative agents are used to reduce pain and anxiety before painful procedures [3,4]. While there are some studies aimed at reducing pain and anxiety during spinal anesthesia and lumbar epidural block applications, studies on TEC are limited [3,5].

The use of midazolam and low-dose ketamine (LDK) has been shown to provide analgesia and anxiolysis during painful or uncomfortable procedures, such as combined spinal–epidural or lumbar epidural alone [5,6]. Ketamine, an N-methyl-D-aspartate antagonist, is a dissociative anesthetic agent and provides effective analgesia [7]. This allows it to be used as an effective pain reliever agent before interventions. Ketamine administered at three different doses (0.15 mg/kg, 0.30 mg/kg, and 0.45 mg/kg) during the performance of the procedures has been shown to decrease patient response, facilitate lumbar puncture, and improve hemodynamic parameters [6]. The side effects of ketamine include hypertension, nausea, agitation, confusion, and hallucinations, although these are dose-dependent and are extremely rare at low doses [8,9,10]. It was observed that the undesirable side effects of ketamine increased at doses of 0.45 mg/kg and above [6]. Ketamine can be used alone or in combination with other drugs [11]. When ketamine is combined with a benzodiazepine, synergistic pain-relieving and sedative effects are observed [12]. Psychological side effects may be rare when ketamine is administered in low doses. Additionally, a combination of ketamine with benzodiazepines also provides a reduction in psychological side effects [8,12]. In combination with midazolam (0.02–0.05 mg/kg), ketamine is used for analgesia and anxiolysis at lower doses (0.5–1 mg/kg) [13]. When given intravenously, it has been shown to be effective within 1 to 2 min (min) [14]. A published study notes that ketamine is associated with the fewest side effects of all agents typically administered for procedural sedation but is used least frequently in this setting [15].

We hypothesized that LDK (0.15 mg/kg) administration before TEC can prevent anxiety and pain during the procedure. The objective of this study was to evaluate the low dose of intravenous (IV) ketamine in reducing pain and anxiety during the TEC procedure.

## 2. Materials and Methods

### 2.1. Study Design and Population

After receiving approval from the clinical research ethics committee of Keçiören Training and Research Hospital (KEAH; ID:238), we included 60 adult patients with ASA physical status I–III who were scheduled for elective thoracic surgery via thoracotomy in this prospective, double-blind and randomized study. This study was registered with the ClinicalTrials.gov international protocol registration and results system under registration number NCT06310850. The patients were given detailed information about the rationale for the study protocol, and informed consent was obtained from all patients. Patients were randomly divided into two groups to receive IV saline (Group P, n = 30) and IV ketamine 0.15 mg/kg (Group K, n = 30), (Figure 1). Randomization was performed using a computerized table of random numbers.

Patients who have chronic pain, bleeding disorder, analgesic drug abuse, liver disease, severe metabolic and endocrine problems, history of allergy to ketamine and local anesthetics, or infection in the intervention area, and patients who refused TEC were excluded from the study. In addition, patients who described acute pain in any part of the body in the preoperative evaluation, whose TEC time was more than five minutes (time from needle entry to catheter placement), and who required more than two Tuohy needle attempts were excluded from the study.

### 2.2. Measurements and Interventions

The baseline vital parameters and anxiety levels of each patient were recorded before premedication. Midazolam 0.03 mg/kg IV was applied for premedication. The visual analog scale (VAS) was used to measure anxiety (VAS-A) and pain (VAS-P) scores [5,16]. For anxiety, on a 100 mm scale, 0 mm was defined as no anxiety, whereas 100 mm was considered unbearable anxiety. A similar scoring method was used to assess pain intensity. Thus, 0 mm was defined as no pain and 100 mm as unbearable pain [5]. Twenty minutes after premedication, patients were transferred to the operating room and received a bolus of 0.15 mg/kg IV ketamine (the provided sample contained equal proportions of two enantiomers, S and R ketamine hydrochloride) or intravenous saline (placebo) three minutes before the TEC placement. The same syringe containing ketamine or the placebo was prepared and administered by an independent nurse anesthetist. Also, the researchers of the study did not know which injector was a placebo or ketamine.

Systolic arterial pressure (SAP), diastolic arterial pressure (DAP), mean arterial pressure (MAP), peripheral oxygen saturation (SpO2), and heart rate (HR) were monitored before premedication (T1), 20 min after premedication (T2), during skin anesthesia (T3), during epidural catheter insertion (T4), and 5 min after epidural catheter placement (T5). VAS-A values were recorded at T1, T3, T4, and T5 periods, and VAS-P levels were noted at T3, T4, and T5 periods. Change (delta-Δ) values were recorded for parameters with different baseline measurements between the groups. The change value at time point T2 was defined as ‘ΔT1–2’, at time point T3 as ‘ΔT1–3’, at time point T4 as ‘ΔT1–4’, and at time point T5 as ‘ΔT1–5’. The side effects of ketamine, including hypertension, nausea, agitation, confusion, and hallucinations, and complications related to TEC were also recorded.

In both groups, the Tuohy needle insertion site was covered with a sterile technique after the skin was cleaned with povidone–iodine. Briefly, 3 mL of 2% prilocaine was used for skin anesthesia, and then an 18-gauge Tuohy needle was inserted through the thoracic 5–6 or 6–7 intervertebral space with median approach using the hanging-drop technique; then, an epidural catheter (Perifix^®^, Braun, Melsungen, Germany) was advanced 3 cm cephalad. Patients were asked to describe whether they remembered the TEC procedure as “I did not remember anything”, “I partially remembered”, or “I remembered the whole procedure”.

The tolerability of the procedure by the patient was defined as “excellent”, “good”, or “moderate”. In addition, the patients were asked whether they would have this procedure performed in the future, and the answers were defined as “yes”, “if necessary”, or “definitely not”.

### 2.3. Outcome

The primary outcome measure was the effect of LDK on the VAS pain score during TEC administration. The secondary outcome measure was the effect of LDK on the VAS anxiety score during the TEC procedure. The third outcome measure was the effect of LDK administered before the TEC procedure on the patient tolerability of the procedure and patient satisfaction.

### 2.4. Sample Size

To calculate the sample size, we based our primary hypothesis on the improvement in the VAS-P score during TEC application with LDK administration compared to the placebo-treated group. As there were no similar clinical studies in the literature, we conducted a pilot study and recruited eight patients from each group. The mean VAS-P change during TEC application was calculated as 35.40 ± 10.40 mm in the placebo group and 29.00 ± 4.50 mm in the ketamine group. Based on the results, the study’s effect size was 0.79, calculated using G*Power© software version 3.1.9.6 (Institute of Experimental Psychology, Heinrich Heine University, Düsseldorf, Germany). The study was designed with 30 patients in each group, with a two-sided (two-tailed) type I error of 0.05, power of 85%, and effect size (d) factor of 0.79.

### 2.5. Data Analysis and Statistics

Data analyses were performed by using SPSS for Windows, version 22.0 (SPSS Inc., Chicago, IL, USA), and figures were drawn with the Jamovi statistical program, version 2.3.21.0 (Sydney, NSW, Australia). Whether the distribution of continuous variables was normal or not was determined by the Kolmogorov–Smirnov test. The Levene test was used for the evaluation of homogeneity of variances. Unless specified otherwise, continuous data were described as median (interquartile range, IQR) and arithmetic mean ± standard deviation (SD) for skewed distributions. Categorical data were described as the number of cases (n, %). The statistical analysis of variables with normal distribution between two independent groups was conducted using Student’s *t*-test. For variables without normal distribution, the Mann–Whitney U test was used for statistical analysis. To compare differences in statistical analyses of non-normally distributed variables between more than two dependent groups, the Friedman test was employed. When the *p*-value obtained from the Friedman test statistic was statistically significant, the Durbin–Conover test was used to determine which group differed from the others. Categorical variables were compared using Pearson’s chi-square test or Fisher’s exact test. A *p*-value < 0.05 was considered significant in all statistical analyses.

## 3. Results

This study compared LDK with a placebo in 60 patients who met the inclusion criteria. Exclusions included five patients whose TEC procedure lasted longer than 5 min, four patients who declined to participate, four patients who reported acute pain, two patients with more than two Tuohy needle insertions, and two patients who did not meet the other inclusion criteria. The patients were randomly assigned to two groups. No significant difference was found between the groups in terms of sex, age, BMI, ASA physical status, the number of attempts, and the duration of TEC (*p* > 0.05), (Table 1).

There were no statistically significant differences between the groups in terms of MAP, HR, and SpO2 values at the measured time points (*p* > 0.05, Figure 2). When comparing the change values of hemodynamic parameters in the time periods measured according to the basal values (T1) between the groups, there were no statistically significant differences in terms of MAP and SpO2 (*p* > 0.05). However, a statistically significant difference was found in terms of HR. No statistically significant difference was found in ΔT1–2 (*p* = 0.244) and ΔT1–3 (*p* = 0.114) values when the changes in HR according to the basal T1 time value were analyzed between the groups. However, a statistically significant difference was found in ΔT1–4 (*p* = 0.011) and ΔT1–5 (*p* = 0.019) values. In Group P, HR increased statistically more in the T4 and T5 time periods compared to Group K.

When VAS-A was compared between the groups, there was no statistically significant difference in T1 (*p* = 0.236) and T3 (*p* = 0.254) time measurements, whereas a statistically significant difference was found in T4 (*p* < 0.001) and T5 (*p* = 0.033) time values (Table 2, Figure 3). When Δ values were analyzed between the groups whose baseline (T1) VAS-A values were not the same, there was a statistically significant difference between the groups only at ΔT1–4 (*p* = 0.010). In Group K, VAS-A was significantly lower during TEC application at the T4 time point. In Group K, there was a mean decrease of 8.97 ± 10.60 mm compared to the basal value during TEC application.

When comparing VAS-P values between the groups, a statistically significant difference was found in the time measurements for T3 and T4 (*p* < 0.001, Table 2, Figure 4). The VAS-P value was significantly lower in Group K during skin anesthesia and TEC application.

When comparing the groups internally, both groups showed a statistically significant difference in VAS-A measurements (*p* < 0.001, Table 2). However, only Group P showed a statistically significant difference in VAS-P measurements (*p* < 0.001, Table 2). During the T3 and T5 times, VAS-A measurements were significantly lower in Group P compared to the T1 time point (*p* < 0.001). However, in Group K, VAS-A measurements were significantly lower at all times compared to T1 (*p* < 0.001). In Group P, both VAS-A and VAS-P increased significantly in the T4 time point. In contrast, there was no statistically significant change in VAS-P measurements at any time in Group K (*p* = 0.099), and the patients’ measurements were more stable.

Although there was no statistically significant difference in terms of complications between the groups, the number of patients with complications during TEC was higher in Group P (*p* > 0.05) (Table 3). We did not observe any ketamine-related side effects.

While the rate of those who did not remember the procedure was statistically higher in Group K, the rate of amnesia in Group P was found to be statistically significantly lower (*p* < 0.001). Also, in Group K, the number of patients who would accept the same procedure in the future was found to be statistically significantly higher (*p* = 0.007) (Table 3).

## 4. Discussion

This study showed that LDK decreased pain and anxiety during the TEC procedure and positively affected the acceptability of the procedure. In addition, fewer side effects were observed in patients who used LDK.

Despite novel regional or peripheric nerve block methods, in thoracic surgery, thoracic epidural anesthesia and analgesia are commonly considered a standard part of general anesthesia and postoperative pain management [17,18]. The majority of the patients imagine these procedures to be unpleasant and painful. Even a comprehensive preoperative interview and explanations about the TEC procedure may not always be sufficient to cope with this anxiety and fear. Anesthesiologists can sometimes overlook the anxiety and fear of the patients about the possible complications of the procedure [3]. Although there are many studies and meta-analyses on the consumption of intravenous ketamine in low doses as an intraoperative anesthetic and postoperative analgesia [19,20], there are only a few studies on pain and anxiety during TEC [2,3,5].

In a trial comparing an intravenous placebo, ketamine 5 mg, and fentanyl 50 mcg administrations before thoracic and lumbar epidural block, Oda et al. [5] found that ketamine and fentanyl had similar anxiety-reducing effects, but pain scores were similar in all three groups. In a study by Mogensen et al. [2], they observed that the predicted pain score before the procedure was significantly lower than the actual pain experienced after the procedure. They also administered additional midazolam (1–2 mg, IV) and/or fentanyl (0.05–0.1 mg, IV) immediately before or during the epidural procedure at the investigators’ discretion as rescue medication in case of patient discomfort. In the present study, midazolam was given IV in both groups 20 min before the procedure. Both anxiety and pain scores were lower in the ketamine group during the epidural procedure compared to the placebo group. In addition, we did not encounter any side effects related to ketamine in our study. These results show us that ketamine reduces pain and anxiety during TEC and that LDK does not lead to ketamine-related psychomimetic effects, in accordance with the literature [8,12].

When ketamine is used at anesthetic doses, it causes transient increases in blood pressure, HR, and cardiac index, presumably secondary to a central sympathomimetic effect [21]. However, these hemodynamic effects are quite limited when ketamine is used at low doses (0.45 mg/kg and above) [6,22]. In our study, we observed an increase in OAB in the placebo group after local anesthetic application to the skin. This increase continued in the ketamine group, although it was limited. While this change suggests that ketamine decreases anxiety and pain due to its sedo-analgesic effect, it also indicates that ketamine at low doses may cause a limited increase in OAB. Additionally, we observed an interesting result in HR. Although HR increased significantly in both groups, the increase was significantly higher in the placebo group. This suggests that the sedo-analgesic effect provided by LDK may suppress the HR response that may develop due to limiting pain and anxiety in patients. Furthermore, the continued increase in HR after the procedure in the placebo group suggests that LDK administration may be a viable option for TEC.

It is well known that ketamine maintains pharyngeal and laryngeal reflexes and respiratory drive at anesthetic doses, which is an advantage in situations where the maintenance of spontaneous ventilation is required [21]. However, high doses of ketamine may increase oral secretions and cause a small increase in the incidence of laryngospasm. As with cardiovascular effects, respiratory effects are rarely observed when ketamine is administered at a low dose [22]. In the present study, we observed stable hemodynamic and respiratory parameters, and we did not confront any desaturation, high blood pressure attacks, or tachycardia. Similar to the results in previous articles, LDK was not related to abnormal physiological effects.

Pain and anxiety during interventional procedures are quite common. Inadequate management of these symptoms will increase complications during the procedure, and the stressful process experienced will cause more intense pain and anxiety in the following procedures. For this reason, comprehensive management before and during the procedure is key to success. First of all, a detailed preoperative interview with the patients is one of the most important stages of this process. In addition, premedication is an effective method [23]. However, sometimes, these practices may not be sufficient to prevent patients’ anxiety and fear [24]. In the present study, more sedation-related amnesia was observed in the ketamine-administered group, and the rate of those who responded positively when asked if they would allow this procedure in the future was quite high in the ketamine group. In addition, the tolerability of the procedure was also found to be higher in the ketamine group, although it was not statistically significant. This result shows that the administration of low-dose preprocedural ketamine provides less anxiety and pain in patients without negatively affecting hemodynamic and respiratory functions, and it provides a positive experience of the procedure by increasing patient comfort.

The effects of LDK on postoperative pain have been studied in many surgical procedures. However, most of these studies have heterogenicity concerning ketamine dose, administration timing, and route. At the same time, it has been stated that, generally, LDK administration is found effective in postoperative pain and analgesic consumption in these trials [19,20,25,26]. Although the effect of LDK, applied before the TEC procedure, on postoperative analgesic consumption and pain scores was not evaluated in this study, we think that preprocedural ketamine administration may contribute to postoperative pain relief in major surgeries such as thoracic surgery.

There are some limitations in this study. First of all, VAS-A was used to assess anxiety. It would have been more effective to measure the anxiety level using state anxiety measurement methods, but since it would not be appropriate to use comprehensive anxiety assessment methods for every patient in clinical practice, VAS-A was used. Second, all epidural blocks were performed by anesthetists with at least one year of TEC experience. However, different durations of TEC experience may affect the pain and anxiety level of the patient. Therefore, it may be appropriate to plan larger studies by anesthetists with different TEC experience levels. Another limitation of the study is that, although low-dose ketamine administration was evaluated, midazolam was administered to both groups because of its anxiolytic effect. This may have limited the study’s effectiveness.

## 5. Conclusions

In conclusion, preprocedural 0.15 mg/kg ketamine was found to be more effective than the placebo in reducing anxiety and pain experienced by patients during the TEC procedure. Ketamine administered before TEC did not cause hemodynamic or respiratory adverse effects, nor any psychological side effects specific to ketamine, since it was used at low doses. In addition, by providing a more comfortable experience during the procedure, LDK may also lead to easy acceptability of later interventions. More powerful studies with larger samples are needed in different doses and combinations of agents.

## Figures and Tables

**Figure 1 medicina-60-00679-f001:**
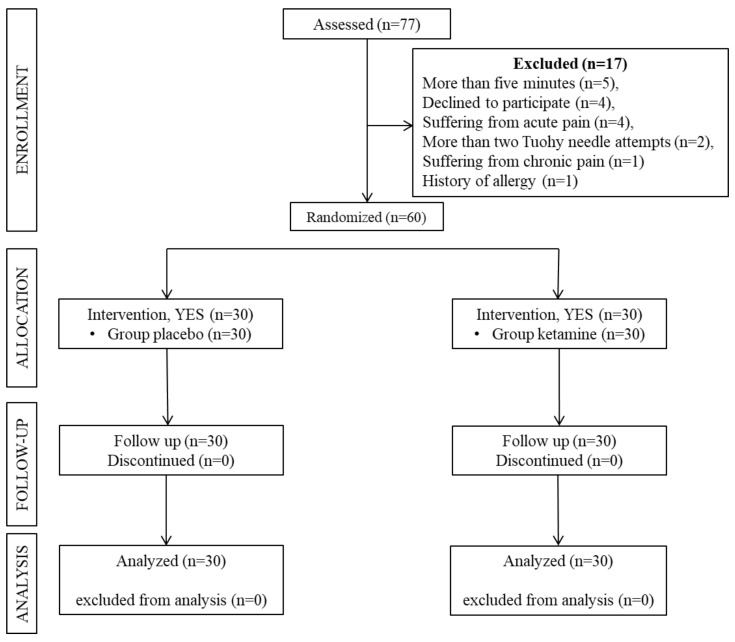
Flow diagram of the participant.

**Figure 2 medicina-60-00679-f002:**
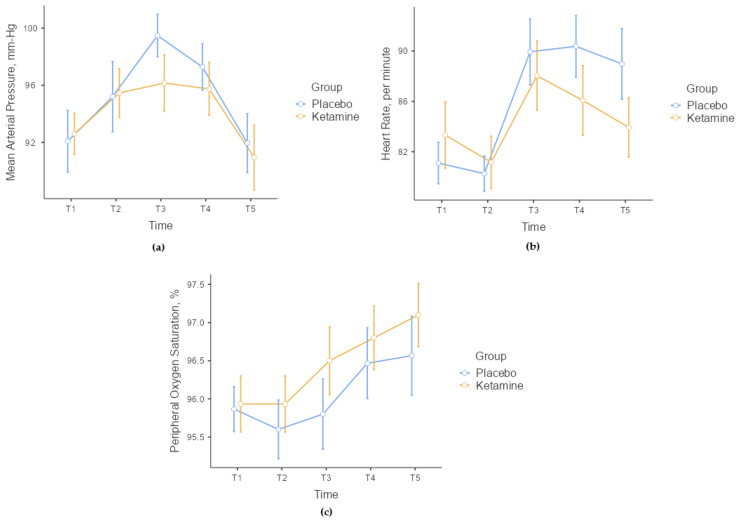
Standard error graphs of mean arterial pressure (**a**), heart rate (**b**), and peripheral oxygen saturation (**c**) according to the time periods measured between groups.

**Figure 3 medicina-60-00679-f003:**
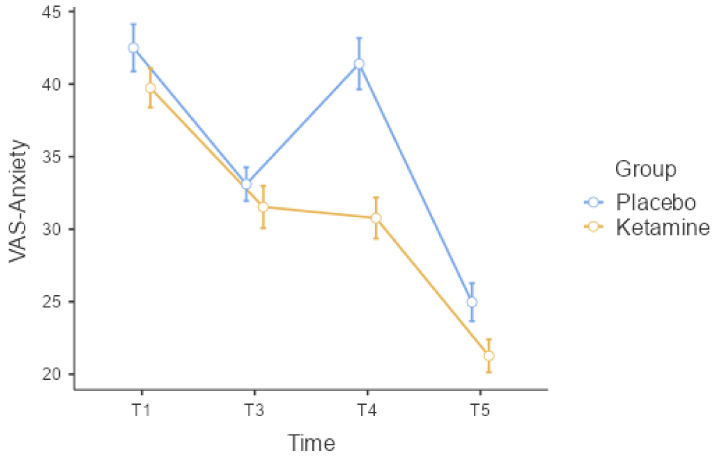
VAS-A error graph according to the time periods measured between groups. VAS-A: visual analog scale—anxiety.

**Figure 4 medicina-60-00679-f004:**
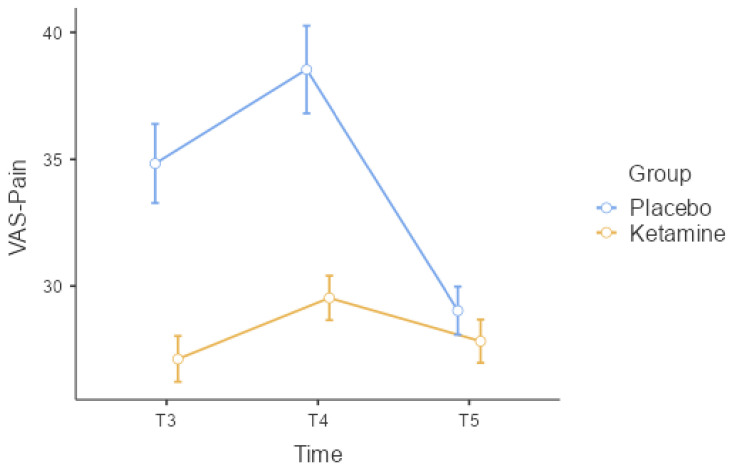
VAS-P error graph according to the time periods measured between groups. VAS-P: visual analog scale—pain.

**Table 1 medicina-60-00679-t001:** Demographic data of patients, number of TEC attempts, and duration of the procedure.

		Group P (n:30)	Group K (n:30)	*p*-Value
Sex, n (%)	Male	18 (60.0)	24 (80.0)	0.091
Age, year, mean ± SD		46.67 ± 14.34	50.13 ± 15.03	0.364
BMI, kg/m^2^, median (IQR)		24.98 (5.02)	25.18 (3.10)	0.988
Duration of procedure, s, median (IQR)		210 (30)	210 (45)	0.917
ASA, n (%)	ASA II	16 (53.3)	13 (43.3)	0.438
	ASA III	14 (46.7)	17 (56.7)	
Number of Attempts, n (%)	1	23 (76.7)	21 (70.0)	0.559
	2	7 (23.3)	9 (30.0)	

Abbreviations: %: percentage, ASA: American Society of Anesthesiologists, BMI: body mass index, IQR: interquartile range, n: number, SD: standard deviation.

**Table 2 medicina-60-00679-t002:** The VAS-A and VAS-P values of patients between the groups and according to baseline values in both groups.

		Group P (n:30)	Group K (n:30)	Independent *p*-Value
VAS-A	T1	41.5 (13.5)	38 (9.75)	0.236
	T3	32 (10.5)	30.5 (8.75)	0.254
	T4	40 (12.5)	31 (7.75)	<0.001
	T5	26 (11.5)	22 (10.3)	0.033
Intra-group *p*-value		<0.001 ^a^	<0.001 ^b^	
VAS-P	T3	35.5 (11)	27 (9.25)	<0.001
	T4	39.5 (12.3)	29.5 (8.25)	<0.001
	T5	28 (7.5)	27 (7)	0.402
Intra-group *p*-value		<0.001 ^c^	0.099	

Statistically significant *p*-values are in bold. The *p* significance value calculated for within-group repeated measurements was called the ‘Intra-group *p*-value’. Significant values for intra-group comparison of VAS-A; ^a^: T1 vs. T3 (*p* < 0.001), T1 vs. T5 (*p* < 0.001), T3 vs. T4 (*p* = 0.002), T3 vs. T5 (*p* = 0.003), and T4 vs. T5 (*p* < 0.001); ^b^: T1 vs. T3 (*p* < 0.001), T1 vs. T4 (*p* < 0.001), T1 vs. T5 (*p* < 0.001), T3 vs. T5 (*p* < 0.001), and T4 vs. T5 (*p* < 0.001). Significant values for intra-group comparison of VAS-P; ^c^: T3 vs. T4 (*p* = 0.019), T4 vs. T5 (*p* < 0.001); VAS-P: visual analog scale—pain; VAS-A: visual analog scale—anxiety; T1: before premedication; T3: during skin anesthesia; T4: during epidural catheter insertion; T5: 5 min after epidural catheter placement.

**Table 3 medicina-60-00679-t003:** Adverse events, level of amnesia, rate of tolerability of procedure, and permission for the same procedure in the future.

		Group P (n:30)		Group K (n:30)		*p*-Value
Adverse events, (n %)	None	22	(73.3)	27	(90.0)	0.274
	Hypotension	5	(16.7)	2	(6.7)	
	Paresthesia	3	(10.0)	1	(3.3)	
Amnesia, (n %)	I did not remember anything	2	(6.7)	15	(50.0)	**<0.001**
	I partially remembered	10	(33.3)	11	(36.7)	
	I remembered the whole procedure	18	(60.0)	4	(13.3)	
Tolerability of procedure, (n %)	Excellent	9	(30.0)	16	(53.3)	0.101
	Good	11	(36.7)	10	(33.3)	
	Moderate	10	(33.3)	4	(13.3)	
Permission for the same procedure in the future, (n %)	Yes	12	(40.0)	23	(76.7)	**0.007**
	If necessary	17	(56.7)	6	(20.0)	
	Definitely not	1	(3.3)	1	(3.3)	

Statistically significant *p*-values are in bold.

## Data Availability

The datasets used and/or analyzed during the current study are available from the corresponding author upon reasonable request.

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
