# Peer review of "The Effect of Preprocedural Low-Dose Ketamine for Pain and Anxiety in Patients during Thoracic Epidural Catheterization"

_medicina, 2024, doi:10.3390/medicina60040679_

Round 1

Reviewer 1 Report

Comments and Suggestions for Authors

Dear authors, thank you for the opportunity to review your manuscript. 

Here are some points that struck me while reading your work:

summary, methods: 

is it an RCT? 

were the patients or staff informed about the treatment? 

can you specify the time interval between the infusion and the intervention? 

subanesthetic dose is a too general description in my opinion. I think the most appropriate definition of 0.15 mg/kg ketamine is “low-dose ketamine” (LDK) or Analgesic-only Ketamine.

summary, results: this part seems confusing to me. Please revise it and describe it more simply. 

Main text: 

Introduction: 

- I would prefer more introduction on the evidence on the use of low-dose ketamine for procedural sedation and/or analgesia instead of on TEC to underline the rationale of your study. 

Material and Methods: 

Is this a single, or a double-blind study? please specify if patients were awere of the arm allocation.

I think there is no need to report both the median and media in case of skewed data. 

How ketamine was administered? In bolus or via quick infusion? (i.e. in 100 ml of 0.9% saline?

Results: 

Table 1: report just male or female sex. 

Table 1: I think there is no need to report the type of statistical analysis you used. 

Table 2: choose one between media (SD) or median (IQR), there is no need to report both. Moreover, use the appropriate test in case of normal or non-normal distribution. 

What does it mean “dependent p value”?

I think Figures 3 and 4 and Table 2 are redundant. Please consider removing the table if data could be reported in a satisfying manner using only figures. 

Discussion: 

I think these two sentences are partially in contrast each other: “These results show us that ketamine reduces pain and anxiety 287 during TEC and that subanesthetic doses of ketamine do not lead to ketamine-related 288 psychomimetic effects, in accordance with the literature [12,16]” and “We think that the absence of ketamine-induced psychomimetic effects may be related to midazolam premedication and low dose ketamine administration”. Despite the use of midazolam may prevent the onset of psychomimetic effects, this is not fully demonstrated, and as you reported, the dose used was demonstrated to be safe from this point of view. Please consider to modify these sentences. 

“When ketamine is used at anesthetic doses, it causes transient increases in blood 292

pressure, HR, and cardiac index, presumably secondary to a central sympathomimetic 293

effect [21]. However, these hemodynamic effects are quite limited when ketamine is used 294 at subanesthetic doses [22]” : I think this is true if you use 0.15 mg/kg of ketamine while using 0.7 mg/kg, which is a sub-anaesthetic dose, may have a very different effect on both hemodynamics and perception. Please, to avoid confusion about the use of this drug, refer to the correct dosage, which is low-dose ketamine or analgesic dose ketamine. 

Conclusion: 

I think a more appropriate conclusion would be that “preprocedural 0.15 mg/kg ketamine demonstrated to be more effective than placebo in reducing anxiety and pain”

Comments on the Quality of English Language

May be improved

Author Response

Dear Editor,

We thank the reviewers for their valuable comments on the manuscript, and we have edited the manuscript to address their concerns. We have highlighted in red all the changed places in the main text. We would like to state that all the reviewers' comments have been taken into account and the necessary corrections have been made.

We hope that the article is now suitable for publication.

Yours sincerely

Ali ALAGÖZ, MD.

University of Health Sciences, Ankara Atatürk Sanatorium Training and Research Hospital, Anesthesiology and Reanimation Clinic, Ankara, Turkey

On behalf of all authors.

Reply to Reviewer-1:

Dear authors, thank you for the opportunity to review your manuscript. 

Here are some points that struck me while reading your work:

summary, methods: 

is it an RCT? 

were the patients or staff informed about the treatment? 

A-1) Thank you for your contribution. As per your suggestion, we have provided a detailed description of the materials and methods used in our study in the abstract. Additionally, we also stated that our study was conducted as a double-blind study.

can you specify the time interval between the infusion and the intervention? 

A-2) In the materials and methods section of our study, we provided a detailed account of administering ketamine 3 minutes prior to the procedure. This information was also included in the summary section.

subanesthetic dose is a too general description in my opinion. I think the most appropriate definition of 0.15 mg/kg ketamine is “low-dose ketamine” (LDK) or Analgesic-only Ketamine.

A-3) The term "subanesthetic dose" has been changed to "low-dose" throughout the article. Thank you for your valuable feedback and contribution. In our literature search, we realised that the term "subanesthetic dose" is used in a very wide dose range. We think that the term "low-dose" is more appropriate for our article.

summary, results: this part seems confusing to me. Please revise it and describe it more simply. 

 A-4) The conclusion of the summary section has been revised in accordance with your valuable suggestion. It has been made clearer and more comprehensible.

Main text: 

Introduction: 

- I would prefer more introduction on the evidence on the use of low-dose ketamine for procedural sedation and/or analgesia instead of on TEC to underline the rationale of your study. 

A-5) The introduction section has been revised to shorten the discussion on thoracic epidural catheter application. Additionally, the literature review on the use of low dose ketamine for sedation, analgesia, and anxiety has been expanded.

Material and Methods: 

Is this a single, or a double-blind study? please specify if patients were awere of the arm allocation.

A-6) Thank you very much for your valuable suggestion. This concept was not directly stated in the design section, but we have revised and added it. We have made an addition to the material method section to clarify that our study is double-blind.

I think there is no need to report both the median and media in case of skewed data. 

A-7) As per your suggestion, we removed mean±SD from the data that did not fit the normal distribution. It is important to note that these values were given in the original manuscript for power analysis in future studies.

How ketamine was administered? In bolus or via quick infusion? (i.e. in 100 ml of 0.9% saline?

A-8) In our study, the application was administered as a bolus and the material was added to the methodology section.

Results: 

Table 1: report just male or female sex. 

Table 1: I think there is no need to report the type of statistical analysis you used. 

Table 2: choose one between media (SD) or median (IQR), there is no need to report both. Moreover, use the appropriate test in case of normal or non-normal distribution. 

A-9) Verilen öneriler doÄŸrultusunda tablo ve açıklamalarda düzeltmeler yapılmıştır.

What does it mean “dependent p value”?

A-10) Thank you for your valuable suggestion. The term 'dependent p value' has been corrected to 'Intra-group p value'. A separate explanation has been added below the table.

I think Figures 3 and 4 and Table 2 are redundant. Please consider removing the table if data could be reported in a satisfying manner using only figures. 

A-11) Thank you for your suggestion. We believe that the table should not be removed as it provides detailed information for the readers. Additionally, Table 2 holds a special place in our study as it presents the significance values of the repeated measurements within the groups.

Discussion: 

I think these two sentences are partially in contrast each other: “These results show us that ketamine reduces pain and anxiety 287 during TEC and that subanesthetic doses of ketamine do not lead to ketamine-related 288 psychomimetic effects, in accordance with the literature [12,16]” and “We think that the absence of ketamine-induced psychomimetic effects may be related to midazolam premedication and low dose ketamine administration”. Despite the use of midazolam may prevent the onset of psychomimetic effects, this is not fully demonstrated, and as you reported, the dose used was demonstrated to be safe from this point of view. Please consider to modify these sentences. 

A-12) The suggestion you provided is appreciated. The prediction about midazolam has been removed from our study.

“When ketamine is used at anesthetic doses, it causes transient increases in blood 292

pressure, HR, and cardiac index, presumably secondary to a central sympathomimetic 293

effect [21]. However, these hemodynamic effects are quite limited when ketamine is used 294 at subanesthetic doses [22]” : I think this is true if you use 0.15 mg/kg of ketamine while using 0.7 mg/kg, which is a sub-anaesthetic dose, may have a very different effect on both hemodynamics and perception. Please, to avoid confusion about the use of this drug, refer to the correct dosage, which is low-dose ketamine or analgesic dose ketamine. 

A-13) Thank you for your valuable suggestion. In line with your suggestion, we have realised that we used the concept incorrectly and corrected it in our entire study. We have stated the concept of sub-anesthetic dose as low dose and used it in this way. We would also like to state that we have added the doses used in the introduction section with the support of the literature and the minimum dose specified in terms of the occurrence of side effects. We have also revised the section you mentioned in the discussion section. We think that the confusion of meaning has been improved in this way. Thank you.

Conclusion: 

I think a more appropriate conclusion would be that “preprocedural 0.15 mg/kg ketamine demonstrated to be more effective than placebo in reducing anxiety and pain”

A-14) Thank you for your valuable suggestion. The relevant section has been edited as specified.

Reviewer 2 Report

Comments and Suggestions for Authors

1) The dosage of 0.1 to 0.3 mg/kg of ketamine has analgesic action. The definition "sub-anesthetic" is incorrect. Furthermore, the "sub-dissociative" definition is highly risky because it refers to higher dosages (0.5-0.9) mg/kg, which are absolutely to be avoided.

2)               In the flow chart on page 3, the reasons for exclusion indicated as "other reasons" for the 2 patients should be better specified

3)               Due to the anxiolytic effect, the administration of midazolam indicates, in part, the effectiveness of the study even if done in both groups.

1) The dosage of 0.1 to 0.3 mg/kg of ketamine has analgesic action. The definition "sub-anesthetic" is incorrect. Furthermore, the "sub-dissociative" definition is highly risky because it refers to higher dosages (0.5-0.9) mg/kg, which are absolutely to be avoided.

2)               In the flow chart on page 3, the reasons for exclusion indicated as "other reasons" for the 2 patients should be better specified

3)               Due to the anxiolytic effect, the administration of midazolam indicates, in part, the effectiveness of the study even if done in both groups.

Author Response

Dear Editor,

We thank the reviewers for their valuable comments on the manuscript, and we have edited the manuscript to address their concerns. We have highlighted in red all the changed places in the main text. We would like to state that all the reviewers' comments have been taken into account and the necessary corrections have been made.

We hope that the article is now suitable for publication.

Yours sincerely

Ali ALAGÖZ, MD.

University of Health Sciences, Ankara Atatürk Sanatorium Training and Research Hospital, Anesthesiology and Reanimation Clinic, Ankara, Turkey

On behalf of all authors.

Reply to Reviewer-2:

  • The dosage of 0.1 to 0.3 mg/kg of ketamine has analgesic action. The definition "sub-anesthetic" is incorrect. Furthermore, the "sub-dissociative" definition is highly risky because it refers to higher dosages (0.5-0.9) mg/kg, which are absolutely to be avoided.

A-1) Thank you for your valuable suggestion. We would like to clarify that we had a misconception and have therefore edited the concept of 'sub-anesthetic' in our study to 'low dose'. This change has made the information clearer and more understandable, eliminating the confusion you mentioned.

  • In the flow chart on page 3, the reasons for exclusion indicated as "other reasons" for the 2 patients should be better specified

A-2) Thank you for your valuable suggestion. The flow diagram for the participant has been edited as specified. The relevant patient data has been detailed.

  • Due to the anxiolytic effect, the administration of midazolam indicates, in part, the effectiveness of the study even if done in both groups.

A-3) Thank you for your valuable suggestion. We have revised the limitations section of our study accordingly. The use of midazolam in conjunction with ketamine is a concept that has been detailed in the introduction section of our study, with supporting literature. We have also added literature-supported information to the introduction section. This concept is also mentioned in the limitations section.
